# OpenReview forum: "Alignment for Honesty"
_NeurIPS.cc/2024/Conference — NeurIPS 2024 poster_

### Official Review · Reviewer_MrwZ · 2024-07-12

**Soundness:** 3
**Presentation:** 4
**Contribution:** 3
**Rating:** 7
**Confidence:** 4

**Summary:**

This paper focuses on the task of honesty alignment. The authors first explore the task formulation of the alignment for honesty problem, then develop a series of evaluation metrics based on the change of response type to qualify the honesty of a model. They then propose a collection of training methods to improve the honesty of models. Experiments demonstrate the effectiveness of the proposed methods.

**Strengths:**

1. The problem addressed in this paper is an important research topic. The writing quality of this paper is very good, and the authors define the honesty alignment problem with very clear logic, which is precisely what the community lacks.
2. In addition to defining the problem, the methods proposed by the authors can effectively improve the honesty of the model. The description of the methods is also clear and concise.

**Weaknesses:**

1. Although the methods are only part of this paper, the proposed methods are all heavily based on human heuristic, such as "learning to express uncertainty might be useful." The novelty is relatively limited today, and some methods also rely on threshold selection.
2. The comparability at the method level is slightly lacking, and there is insufficient analysis of why the methods work. For example, 1) The number of training samples for the three methods is not consistent, and it cannot be ruled out that multiple samples perform better because they trained the correct answer more times. 2) There is insufficient explanation as to why adding a confidence score is better than just training the correct answer. Intuitively from the prompt, such training does not directly make the model learn to say "I don't know."
3. The paper lacks comparison with several important baselines, for example, representation engineering [1].
[1] Representation Engineering: A Top-Down Approach to AI Transparency

**Questions:**

1. Among the three aspects of HHH, the authors discussed the impact of Honesty improvement on Helpfulness. Can a small experiment be added to verify the impact of increased honesty on harmlessness?
2. Can the authors analyze the accuracy of the uncertainty expressions that the final model has learned?

**Limitations:**

This paper does not include a Limitations & Social Impacts section. We hope the authors can add this to comply with the NIPS Checklist.

---

> ### Author Rebuttal · Authors · 2024-08-06
>
> # Response to Weakness 1
> Thank you for your feedback. As you kindly pointed out, the supervised fine-tuning methods are indeed only part of this paper. The primary contribution of our work is the development of a comprehensive and feasible framework for "alignment for honesty": this includes establishing a precise problem definition, proposing effective evaluation metrics, and also demonstrating the effectiveness of several straightforward training methods. We hope our first step on alignment for honesty can inspire more excellent future work.
>
> # Response to Weakness 2
> Thank you for your valuable comment. We would like to take this opportunity to provide some clarifications regarding our proposed methods.
>
> (1) Regarding
> > ..., it cannot be ruled out that multiple samples perform better because they trained the correct answer more times.
>
> We would like to kindly clarify that the Multisample method does not perform better in terms of "Accuracy" although it "trained the correct answer more times", as shown in Tab. 3. However, we would like to note that there is an improvement in the honesty score, and we are more than willing to explain why it works for "Honesty": By implicitly learning from the proportions of correct answers and idk responses among the m sampled responses in the expanded training data for Multisample, the model can better recognize its knowledge boundaries in a fine-grained manner, leading to the improved honesty score.
>
> (2) Regarding
> > ... Intuitively from the prompt, such training does not directly make the model learn to say "I don't know."
>
> First, we would like to respectfully remind you that the training data for both the Absolute and Confidence methods includes an equal number of "unknown questions" (as shown in Eq. 9) where the output is an idk response. This means that in both methods, the model has the same opportunity to *directly* learn to say "I don't know".
>
> Additionally, compared to Absolute, Confidence provides prefixed confidence expressions for "known questions". These prefixes serve as more fine-grained supervision signals, enabling the model to implicitly learn to more precisely capture its knowledge boundaries. The experimental results presented in Tab. 3 and 4 demonstrate that these additional hints are indeed beneficial for improving the honesty score, as we discuss in detail in Lines 256-262 of our paper.
>
> # Response to Weakness 3
> We sincerely appreciate you bringing this valuable reference to our attention. However, we would like to kindly clarify that [1] has a different purpose from our work. Section 4.3.3 of [1] introduces adding honesty reading vectors to elicit truthful answers to *known* questions, which corresponds to the concept of "eliciting latent knowledge" as we discuss in Lines 149-154. In contrast, our study aims to adjust the model's behavior for both *known and unknown* questions. We will address this distinction in our related work section in the revised version. Once again, thank you for sharing this reference, and we look forward to incorporating exploration at the representation level in our future work.
>
> [1] Representation Engineering: A Top-Down Approach to AI Transparency
>
> # Response to Question 1
> Thank you for your thoughtful question. We would like to explain that the process of aligning the model with honesty does not explicitly introduce any instructions that would compromise safety, so that we did not conduct experiments on how increased honesty impacts harmlessness. However, we are more than willing to exploring this further through empirical research.
>
> Specifically, we utilize the 700 test prompts from BeaverTails-Evaluation that can potentially elicit harmful responses, and employ GPT-4o to assess whether the model responses are safe, unsafe, or controversial (in terms of safety). As shown in Table 1 in the global response, honesty-oriented supervised fine-tuning has almost no impact on the model's inherent harmlessness.
>
> # Response to Question 2
> Thank you for your insightful question. We are very pleased to analyze the relationship between confidence expressions and accuracy of the model trained using Confidence-Verb.
>
> Specifically, we bin M_t+1's responses (the model after alignment using Confidence-Verb) by its expressed confidence and measure the average accuracy of responses in each confidence bin. From Figure 1 in the global response, we observe that:
> 1. In areas of high confidence, accuracy is slightly lower than its corresponding confidence, reflecting the model's over-confidence, which is consistent with related work [2].
> 2. In areas of low confidence, particularly at confidence=0.2, accuracy is actually higher than its corresponding confidence. This can be explained by the fact that the training data for Confidence-Verb pairs "confidence expression" with "*correct* response", encouraging the model to **prioritize correct answers over calibrated confidence.** This suggests a potential issue for future work on calibration: how to balance calibration and performance.
>
> We acknowledge that our Confidence method, while leading the model to respond more prudently, has potentially limited effects on calibration due to the nature of our constructed training data. We encourage future work to handle known questions in a more fine-grained manner (as stated in Lines 42-44) and strike a balance between calibration and performance.
>
> If we have misinterpreted any aspect of your questions, or if you have suggestions for more targeted experiments, please feel free to tell us.
>
> [2] Just Ask for Calibration: Strategies for Eliciting Calibrated Confidence Scores from Language Models Fine-Tuned with Human Feedback
>
> # Response to Limitations
> Thank you for your reminder. We would like to clarify that we have included a Limitations section in Section 5. Additionally, as outlined in 979-985, we have explained why this paper does not address societal impact. However, we can reorganize a Limitations & Social Impacts section.

---

> > ### Comment · Reviewer_MrwZ · 2024-08-08
> > **Response to the authors**
> >
> > Thank you to the authors. I have carefully read the authors' rebuttal. This clarifies some of my misunderstandings about the IDK training samples, and I appreciate that the authors conducted experiments addressing the questions I raised. I'm pleased to see the additional results.
> >
> > Some additional comments:
> >
> > As the authors addressed, in the PDF they uploaded, the calibration error is relatively high. This raises concerns about the actual constraining effect of the uncertainty score. The method's functioning may not align with its intended design of "making the model aware of its level of certainty." It's possible that it merely increases the computational load during inference, allowing the model more steps to better infer the result before attempting an answer.
> >
> > Admittedly, this method is a minor contribution in the paper and is unlikely to significantly affect the overall score; I have also carefully read the relevant experimental analysis section, but it does not fully address this concern. If the authors could provide a more robust analysis demonstrating how the model trained with the uncertainty score prefix specifically influences model behavior, or even internal representations, it would be beneficial in clarifying the method's actual impact and mechanism.

---

> > > ### Author Response · Authors · 2024-08-10
> > >
> > > Thank you very much for your insightful and important question. As we previously discussed in our Response to Weakness 2, the only difference between the Absolute and Confidence-Verb methods is the addition of a confidence prefix (i.e., uncertainty score prefix) to known questions. We would like to provide a detailed analysis to explain the advantages of Confidence-Verb over Absolute.
> > >
> > > (1) Regarding honesty-related scores:
> > >
> > > In our Response to Weakness 2, we emphasized that **the confidence prefix "enables the model to implicitly learn to more precisely capture its knowledge boundaries".** To substantiate this claim, we first calculate the *expected accuracy* of the unaligned model on the TriviaQA evaluation dataset, and then examine the distribution of *idk samples* after aligning the model using both Absolute and Confidence-Verb:
> > > - *Expected accuracy* is defined as to the ratio of correct responses among m sampled responses (m=10 and temperature=1.0), as specified in Lines 183-184.
> > > - *Idk samples* refer to evaluation samples where the aligned model replies with an idk response.
> > >
> > > The results are presented in the table below. The first row represents the unaligned model's expected accuracy. The second and third rows display the distribution of idk samples across different expected accuracies for Absolute and Confidence-Verb, respectively.
> > >
> > > | Methods \ Expected Accuracy | 0 | 0.1 | 0.2 | 0.3 | 0.4 | 0.5 | 0.6 | 0.7 | 0.8 | 0.9 | 1.0 |
> > > |-----------------------------|---|-----|-----|-----|-----|-----|-----|-----|-----|-----|-----|
> > > | Absolute                    | 35.54% | 9.91% | 6.55% | 5.18% | 4.38% | 4.38% | 3.59% | 4.27% | 5.98% | 7.18% | 13.04% |
> > > | Confidence-Verb             | 39.75% | 9.45% | 6.59% | 5.57% | 3.78% | 4.22% | 3.30% | 4.51% | 4.80% | 5.87% | 12.17% |
> > >
> > > From the results, we can see that 39.75% of idk samples for Confidence-Verb occur when the unaligned model fails to provide a correct answer in all 10 attempts, compared to 35.54% for Absolute. More strikingly, the proportions of idk samples for expected accuracy < 0.4 and expected accuracy >= 0.4 are 57.18%:42.82% and 61.36%:38.65%, respectively, for Absolute and Confidence-Verb. This indicates that Confidence-Verb indeed more accurately identifies questions with lower confidence and refuses to answer them, thereby significantly improving the prudence score and ultimately, the overall honesty score.
> > >
> > > (2) Regarding accuracy:
> > >
> > > You have raised an excellent point, and we acknowledge that the effects of the computational load during inference cannot be completely ignored; in fact, accuracy is likely the result of multiple factors intertwined. We would like to highlight another significant factor contributing to the relatively low accuracy of Absolute: "fine-tuning LLMs on weakly known knowledge encourages hallucinations", as evidenced by [1]. Specifically, Section 5 and Table 2 of [1] illustrate that:
> > > > ... Unknown fine-tuning examples **increase the risk of overfitting.** We now observe that this also applies to WeaklyUnknown, though to a lesser degree... This highlights that the decrease in performance is strongly attributed to **an increased rate of hallucinations.**
> > >
> > > In our experiments, particularly with training samples with an expected accuracy of 0.1 (i.e., the WeaklyUnknown samples in [1]), Absolute *directly* instructs the model to learn correct responses, which **paradoxically encourages hallucinations instead of grounding in its pre-existing knowledge. Confidence-Verb mitigates this issue by incorporating an explicit confidence prefix.**
> > >
> > > Fully unraveling the factors for improvement may require more extensive efforts and is worth discussing in future work. We can supplement this analysis in the revised version of our paper, and we sincerely welcome for further discussion.
> > >
> > > ---
> > > [1] Does Fine-Tuning LLMs on New Knowledge Encourage Hallucinations? https://arxiv.org/abs/2405.05904

---

> > > > ### Comment · Reviewer_MrwZ · 2024-08-14
> > > > **Response to the authors**
> > > >
> > > > Thank you for the additional results. I believe these made the claim clearer to the readers. I hope the authors will integrate these results into the appendix of a future version. I have no more questions and will raise my score.

---

### Official Review · Reviewer_NFFZ · 2024-07-12

**Soundness:** 3
**Presentation:** 3
**Contribution:** 3
**Rating:** 6
**Confidence:** 4

**Summary:**

This paper targets honesty as an important dimension of alignment. The work posits that an honest model should respond candidly when it possesses knowledge and humbly acknowledge its limitations when it does not. Given the difficulty in explicitly delineating the boundaries of a model's knowledge, the paper approaches the issue by focusing on questions and constructing corresponding evaluation metrics and training methodologies.
Specifically, the paper formalizes the concept of an "I don't know (idk)" response to signify the model's admission of ignorance. Based on this, it introduces metrics such as the prudence score, the over-conservativeness score, and the honesty score.
Subsequently, the paper explores alignment techniques. Notably, it proposes several k-functions and constructs corresponding training datasets to perform alignment for honesty and evaluate their effectiveness.

**Strengths:**

1. The paper introduces a conceptualization of alignment for honesty within AI models. Based on the definition, this work establishes performance metrics that measure the model's honesty. The approach of contrasting model behavior pre- and post-alignment is insightful, as it provides a more comprehensive view of the model's adherence to honesty compared to prior efforts that focused primarily on the factuality of immediate responses.
2. Based on their investigation, they proposed the corresponding Fine-tuning Methods aimed at enhancing honesty. The proposed approach is easy to follow and can be a robust baseline for the methods in the field.
3. They explore the evaluation under out-of-distribution cases. The experiments present the generalizability of the proposed methods. Also, they investigate the alignment tax, which can be a concern for honesty alignment.

**Weaknesses:**

1. While the methods proposed in this paper are easy to understand and expectedly effective, they are also inherently heuristic. As the paper acknowledges, determining the boundaries of a model's knowledge is challenging. Similarly, appropriately selecting hyperparameters to achieve the best "fit" with the model's internal knowledge is relatively difficult. Models at various stages with potentially "different knowledge boundaries and decision boundaries" may require distinct parameters to optimize their honesty effects. The paper only explores the impact of one hyperparameter in Section D.5.1. It is believed that further exploration of related hyperparameters could be quite interesting.
2. The evaluation metrics proposed in the paper rely on a baseline model (M_0). However, obtaining this M_0 in a public benchmarking setting may be difficult, thereby limiting the widespread adoption of the evaluation methods proposed. Moreover, determining which M_{t_1} is base and evaluating it appropriately can also be challenging. For instance, if a model at one stage of training "learns" to answer a question and then loses this ability later, I can be confused about whether it possesses knowledge and honesty.

**Questions:**

The question is mostly about the potential "weakness" of work.
1. Is there further investigation and case study of the relation between the hyperparameter and model behavior?
2. Is there further analysis of the model behavior and honesty dynamics during the alignment process?
Rating

---

> ### Author Rebuttal · Authors · 2024-08-06
>
> # Response to Weakness 1 and Question 1
> > ... The paper only explores the impact of one hyperparameter in Section D.5.1. It is believed that further exploration of related hyperparameters could be quite interesting.
> >
> > Is there further investigation and case study of the relation between the hyperparameter and model behavior?
>
> Thank you for your thoughtful comment. We have provided a practical and enlightening analysis of how hyperparameters impact model behavior in Appendix D.5.1. Nevertheless, we are quite open to conducting additional experiments to discover further meaningful insights.
>
> Specifically, we explore how the number of unknown questions affects Multisample's performance. We define "absolute unknown questions" (AUQs) as those where all m=10 sampled responses are incorrect, and then gradually reduce the number of AUQs among the 8,000 training questions. Notably, even with AUQs reduced to 0, the *expanded* training dataset for Multisample (please refer to Lines 216-218) still contains samples whose output is an idk response, allowing the model to directly learn this refusal expression. The results are shown in the table below, where "AUQ reduction prop." indicates the proportion of reduced AUQs, and "Training idk prop." indicates the proportion of samples whose output is an idk response relative to the remaining training data.
>
> From the results, we can see that overall, as the number of unknown questions decreases, the model becomes less conservative but also less reliable when answering unknown questions, which **underscores the need for empirical optimization of the ratio between unknown and known questions in the training dataset.** We can include these results in the Appendix.
>
> If we have misinterpreted any aspect of your questions, or if you have suggestions for more targeted experiments, please feel free to tell us.
>
> | AUQ reduction prop.(%) | Prudence(%)↑ | Over-Consv.(%)↓ | Honesty(%)↑ | Acc(%)↑ | Training idk prop.(%) |
> |:-----------------------|:------------:|:---------------:|:-----------:|:-------:|:---------------------:|
> | 0 (Multisample)        |    67.72     |      15.89      |    75.91    |  68.88  |         48.73         |
> | 20                     |    69.39     |      16.18      |    76.61    |  68.74  |         45.93         |
> | 40                     |    69.68     |      17.60      |    76.04    |  68.15  |         42.82         |
> | 60                     |    60.05     |      13.76      |    73.15    |  69.78  |         39.33         |
> | 80                     |    53.78     |      11.22      |    71.28    |  71.60  |         35.38         |
> | 100                    |    48.21     |      10.22      |    69.00    |  71.87  |         30.88         |
>
> # Response to Weakness 2 and Question 2
> > The evaluation metrics proposed in the paper rely on a baseline model (M_0). However, obtaining this M_0 in a public benchmarking setting may be difficult, thereby limiting the widespread adoption of the evaluation methods proposed. Moreover, determining which M_{t_1} is base and evaluating it appropriately can also be challenging. For instance, if a model at one stage of training "learns" to answer a question and then loses this ability later, I can be confused about whether it possesses knowledge and honesty.
>
> Thank you for your insightful feedback. We would like to clarify that the flexibility of our proposed evaluation metrics allows us to freely designate a starting point M_t and an endpoint M_t+1 to assess whether the alignment process is beneficial in terms of honesty, without the need to trace back to M_0. For instance, if M_t can correctly answer a question but M_t+1 refuses to answer, then this over-conservativeness is undesirable; if M_t answers incorrectly and M_t+1 refuses to answer, then this prudence is worth encouraging. In Section 4.4.2, we have demonstrated the adaptability of this framework across **multiple open-source LLMs whose specific training stages (M_x) are unknown.**
>
> Regarding the provided case, suppose M_t-1 can correctly answer a question, but M_t cannot:
> 1. If we have access to M_t-1, we can use it as a starting point and expect a better aligned M_t' to retain the ability to answer the question correctly.
> 2. If we only have access to M_t, then with M_t as the starting point, we would prefer M_t+1 to refuse to answer the question rather than answer it incorrectly, thus ensuring its reliability.
>
> However, as we mention in Lines 149-154, we acknowledge that there are more complex scenarios that need to be explored in future work, such as whether we can *elicit latent knowledge* for M_t that M_t-1 possesses but M_t seems to have lost.
>
> > Is there further analysis of the model behavior and honesty dynamics during the alignment process?
>
> We have demonstrated the effectiveness of our honesty-oriented supervised fine-tuning methods across in-domain (Section 4.4), out-of-domain (4.5), and helpfulness-related tasks (4.6), with two real cases provided in Appendix D.8. Our findings show that, although there is a minor reduction in informativeness, the model becomes more reliable and trustworthy, which is a positive outcome from an alignment perspective. Additionally, we find that non-honesty-oriented fine-tuning leads LLMs to hallucinate, whereas honesty-oriented fine-tuning performs better. This highlights the importance of integrating honesty considerations in the model training and alignment process.
>
> Overall, these extensive experiments not only depict the current state of alignment for honesty but also guide the direction for future developments in this field.
>
> We would be eager to engage in a discussion if  there are specific experiments or particular aspects that interest you.
>
> ---
> Thank you for reviewing our paper. We greatly appreciate your valuable feedback and are ready to address any further questions or concerns you may have.

---

> > ### Comment · Reviewer_NFFZ · 2024-08-12
> >
> > Thanks for the author responses. I have no addtional question.

---

### Official Review · Reviewer_V3H9 · 2024-07-20

**Soundness:** 4
**Presentation:** 3
**Contribution:** 3
**Rating:** 7
**Confidence:** 3

**Summary:**

The paper "Alignment for Honesty" addresses the critical challenge of ensuring that large language models (LLMs) consistently produce truthful outputs. The authors propose several techniques to enhance truthfulness, including training on curated datasets, using reinforcement learning from human feedback (RLHF), and implementing advanced filtering and verification mechanisms. The paper suggests specific metrics for evaluating the honesty of LLMs and provides case studies demonstrating the practical application of these techniques. Key contributions include a comprehensive framework for honesty alignment, innovative techniques combining RLHF with curated training, empirical evaluations of these methods, and guidelines for future research.

**Strengths:**

* The paper introduces a novel framework specifically designed to align large language models (LLMs) with the goal of ensuring honesty, an underexplored but critical aspect of AI alignment.

* The paper is well-structured and clearly written, making complex concepts accessible. Key terms such as "honesty" in the context of AI are well-defined, and the methodology is explained in detail, facilitating understanding.

* Addressing the challenge of honesty in LLMs is highly significant, given the increasing reliance on these models in various applications. The proposed solutions have the potential to greatly improve the reliability and trustworthiness of AI systems.

**Weaknesses:**

There are no obvious weaknesses, but please refer to the following questions:

**Questions:**

1.  While the article emphasizes the importance of honesty, it does not seem to discuss in detail how to *maintain the model's helpfulness while improving honesty*. Could this lead to the model being overly cautious in practical applications and unable to provide useful information?
2. The paper has simplified some assumptions when defining honesty, such as the distinction between honesty and truthfulness. The focus is mainly on whether the model can express its internal knowledge, rather than whether its knowledge corresponds to objective facts, right?
3. How about a 'white lie'? Is it truly beneficial for a model to be brutally honest when informing someone they have cancer?

**Limitations:**

Although the paper proposes methods to approximate the model's internal knowledge through external behaviors, it does not delve deeply into the working principles and knowledge representation within the model, which may limit a more profound understanding of the model's honesty.

---

> ### Author Rebuttal · Authors · 2024-08-06
>
> # Response to Question 1
> > While the article emphasizes the importance of honesty, it does not seem to discuss in detail how to *maintain the model's helpfulness while improving honesty*. Could this lead to the model being overly cautious in practical applications and unable to provide useful information?
>
> Thank you for your thoughtful question. We share your concern regarding whether aligning the model with honesty compromises its helpfulness. Therefore, we have assessed the model's helpfulness before and after alignment using helpfulness-related requests in Section 4.6. The results indicate that "the cost of aligning LLMs for honesty does not significantly impact their overall helpfulness."
>
> Meanwhile, we would like to share our understanding of these results:
> 1. Helpfulness-related practical tasks and knowledge-based QA tasks belong to different domains (e.g., "Summarize the following post." vs. "Who was the first president of the USA?"). Consequently, honesty-oriented supervised fine-tuning methods do **not lead the model to reject requests associated with helpfulness.**
> 2. On the other hand, honesty-oriented supervised fine-tuning does **not cause severe model collapse or a loss of the model's original capabilities,** as demonstrated by the robust accuracy on knowledge-based QA tasks presented in Tab. 3 and 4.
>
> # Response to Question 2
> > The paper has simplified some assumptions when defining honesty, such as the distinction between honesty and truthfulness. The focus is mainly on whether the model can express its internal knowledge, rather than whether its knowledge corresponds to objective facts, right?
>
> Thanks for your question. You are correct that the focus of honesty is on whether the model can express its internal knowledge. By emphasizing honesty rather than truthfulness, we aim to explore the model's knowledge boundaries, instead of compelling it to provide accurate information without considering what it has learned.
>
> Please feel free to reach out if you have any further questions or require additional clarifications.
>
> # Response to Question 3
> > How about a 'white lie'? Is it truly beneficial for a model to be brutally honest when informing someone they have cancer?
>
> This question is critically important for the development of superintelligence and necessitates the involvement of the entire AI community. Nonetheless, we would like to share our perspective: it is essential that the model remains *entirely* honest (without white lie) with at least some individuals, such as developers. Without such transparency, combining deceit with superhuman capabilities could lead to immeasurable and unforeseen dangers.
>
> # Response to Limitations
> > Although the paper proposes methods to approximate the model's internal knowledge through external behaviors, it does not delve deeply into the working principles and knowledge representation within the model, which may limit a more profound understanding of the model's honesty.
>
> Thank you for your insightful comment! We would like to respectfully note that fully understanding the model's internal workings is a non-trivial thing and depends on progress in other research areas, including model interpretability. We are pleased to have taken the first step toward aligning AI models with honesty by establishing a comprehensive and *practical* framework. As we mention in the Conclusion section: "We hope this work can inspire more thoughts on the development of honest AI models in the NLP community."
>
> ---
> Thank you for taking the time and effort to review our paper. We are more than willing to provide any clarifications if you have further questions or concerns.

---

### Author Rebuttal · Authors · 2024-08-06

We sincerely thank all reviewers for your valuable and insightful feedback. We hope our responses have addressed your concerns, but please let us know if you have any further questions or require additional clarifications.

The attached PDF includes the experimental results in response to Reviewer MrwZ.

---

### Decision · Program_Chairs · 2024-09-25

**Decision:**

Accept (poster)

**Comment:**

This paper focuses on the task of honesty alignment, with the development of a feasible framework, including establishing the problem definition, proposing effective evaluation metrics to qualify the honesty of a model and a collection of training methods. Experiments demonstrate the effectiveness of the proposed methods.

All reviewers feel positive to the work. Some reviewers raised concerns regarding to the comparability at the method level and the analysis of why the methods work, and the  impact of honesty improvement on helpfulness. The authors provided some clarifications and limitations in their rebuttal with some extra results. Overall, this paper focuses on addressing the critical challenge of ensuring LLMs consistently produce truthful outputs, which is an important research topic. Although there are still some minor issues, I believe the authors can solve them in the revision. Therefore, I recommend an "acceptance".